# The Clinical Profile of Cat-Scratch Disease’s Neuro-Ophthalmological Effects

**DOI:** 10.3390/brainsci12020217

**Published:** 2022-02-04

**Authors:** Sanda Jurja, Alina Zorina Stroe, Mihaela Butcaru Pundiche, Silviu Docu Axelerad, Garofita Mateescu, Alexandru Octavian Micu, Raducu Popescu, Antoanela Oltean, Any Docu Axelerad

**Affiliations:** 1Department of Ophthalmology, Faculty of Medicine, ‘Ovidius’ University of Constanta, 900527 Constanta, Romania; jurjasanda@yahoo.com; 2County Emergency Clinical Hospital “Sf. Apostol Andrei”, Tomis Street, nr. 145, 900591 Constanta, Romania; mihaelapundiche@yahoo.com (M.B.P.); docuaxi@yahoo.com (A.D.A.); 3Department of Neurology, General Medicine Faculty, Ovidius University, 900470 Constanta, Romania; 4Surgery Department, Faculty of General Medicine, ‘Ovidius’ University of Constanta, 900470 Constanta, Romania; 5Faculty of General Medicine, Vasile Goldis University, 317046 Arad, Romania; docu.silviu@yahoo.com; 6Morphology Department, Faculty of Medicine, University of Medicine and Pharmacy, 200349 Craiova, Romania; garofita.mateescu@umfcv.ro; 7Department of Economic Engineering in Transports, Maritime University of Constanta, Str. Mircea cel Bătrân, 104, 900663 Constanta, Romania; micu.alexandru@gmail.com; 8Physical Education, Sport and Kinetotherapy Department, ‘Ovidius’ University of Constanta, 900470 Constanta, Romania; raducu.popescu22@gmail.com (R.P.); olteantoanela@gmail.com (A.O.)

**Keywords:** cat scratch disease, *Bartonella henselae*, neuro-ophthalmology, ocular, neurological

## Abstract

Cat-scratch disease is an illness caused by *Bartonella henselae* that occurs as a result of contact with an infected kitten or dog, such as a bite or scratch. It is more prevalent in children and young adults, as well as immunocompromised individuals. There are limited publications examining the features of CSD in patients. As such, the purpose of this research was to assess the clinical neuro-ophthalmological consequences of CSD reported in the literature. Among the ophthalmologic disorders caused by cat-scratch disease in humans, Parinaud oculoglandular syndrome, uveitis, vitritis, retinitis, retinochoroiditis and optic neuritis are the most prevalent. The neurological disorders caused by cat-scratch disease in humans include encephalopathy, transverse myelitis, radiculitis, and cerebellar ataxia. The current review addresses the neuro-ophthalmological clinical manifestations of cat-scratch disease, as described in papers published over the last four decades (1980–2022). All the data gathered were obtained from PubMed, Medline and Google Scholar. The current descriptive review summarizes the most-often-encountered clinical symptomatology in instances of cat-scratch disease with neurological and ocular invasion. Thus, the purpose of this review is to increase knowledge of cat-scratch disease’s neuro-ophthalmological manifestations.

## 1. Introduction

Cat-scratch disease (CSD), caused by *Bartonella henselae*, is a global anthropozoonosis that causes major public health issues. As CSD is not a reportable illness in many countries, determining the actual incidence is challenging. *B. henselae* and CSD infections may occur everywhere. The prevalence of infection, according to the literature, is much higher than clinically acknowledged, as shown by serological tests.

Using appropriate search criteria, our investigation retrieved papers from the PubMed and Medline databases. Relevant English-language articles from 1980 to 2022 were identified via a search using the phrases “cat scratch disease” and “*Bartonella henselae* infection” combined with “neurologic symptoms”, “ophthalmologic symptoms”, “neurology”, or “ophthalmology”. Another search combined the terms “cat scratch disease” and “*Bartonella henselae* infection” and the terms “Parinaud oculoglandular syndrome”, “uveitis”, ”vitritis”, ”retinitis”, “retinochoroiditis”, “optic neuritis”, “encephalopathy”, “myelitis”, and “radiculitis”. One hundred articles were selected from the databases. The inclusion criteria were articles from the scientific literature related to the cat-scratch disease, especially related to the subjects of neurology and ophthalmology. The exclusion criteria were animal research and illnesses other than cat-scratch disease. The papers retrieved via the search were examined, and the pertinent information is presented in this work as a narrative review.

## 2. Transmission Agent

The causative agent of cat-scratch disease is represented by the genus *Bartonella* Gram-negative bacilli [1,2,3,4]. The principal species involved in cat-scratch disease is *Bartonella henselae*, previously referred to as *Rochalimaea henselae*, but in the literature, it has also been described as *Bartonella clarridgeiae*, as an etiologic agent (Table 1) [5]. Cats, particularly younger and stray ones, represent the elemental reservoir of each *Bartonella* species [6]. The vector involved in the cat-to-cat transmission of the disease is *Ctenocephalides felis*.

Transmission to humans happens mainly through a cat’s scratch or bite, and less ordinarily, through contact with the mucous membrane. Outdoor cats and flea-infected cats have the greatest infection rates. Compared to adult cats, kittens have greater levels of bacteremia, making them more effective carriers. The bacterium has also been found in dogs, but the clinical relevance of dogs carrying the bacterium in human infections is unknown.

Cats and dogs are usually asymptomatic carriers. Diseases of cats infected with *B. henselae* can include anemia and diaphragmatic myositis, and markers and manifestations of infection/inflammation, including eosinophilia, fever, hyperglobulinemia, lethargy, and lymphadenomegaly, are also present. In addition, mild neurological signs may be present, as well as cardiac manifestations such as pyogranulomatous myocarditis, endocarditis, endomyocarditis, and endocardial fibrosis complex. Ocular manifestations include uveitis, conjunctivitis, keratitis, and corneal ulcers [7].

The manifestations of the infection of *B. henselae* in dogs include lymphadenomegaly, endocarditis, eosinophilia, epistaxis, fever, and granulomatous inflammation [6,7]. Glucidic metabolism can be affected by hyperinsulinemic and hypoglycemia syndrome. The hepatic manifestations include granulomatous hepatitis and peliosis hepatis [6,7]. Additionally, vasoproliferative lesions can be present.

The primary approach to prevention would be to avoid contact with cats. However, cat owners can adopt the following highly effective preventative techniques for flea management in cats: washing hands thoroughly after interaction with a cat; avoiding contact with stray cats—especially kittens; avoiding cat licks, particularly in the mouth, nose, and ophthalmic region; and generally maintaining a pleasant relationship with the cat, without scratching or biting.

A key strategy for preventing *B. henselae* infection in cats is to manage fleas and other ectoparasites. Additionally, keeping cats’ claws short is beneficial. With regard to the surroundings, owners should maintain cleanliness and be vigilant about pest management. Additionally, minimizing a cat’s interaction with other or stray cats and potentially infected animals may be beneficial. Owners should also schedule routine veterinarian health examinations. There is currently no vaccine available to prevent infection with *Bartonella*. 

Ectoparasites such as fleas, ticks, and mites are often discovered on cats and dogs and are capable of harboring *Bartonella* spp. Lice, fleas, and sandflies have been identified as vectors of five *Bartonella* species including *Bartonella henselae*. Numerous mites, keds, and biting flies, as well as, probably, ticks, are now considered possible or potential vectors of *Bartonella* transmission [7]. Flea excrement is infectious for a long time and has been linked to *B. henselae* infection through the direct inoculation of exposed wounds or mucous membranes such as the conjunctiva [7].

One means for the infection of the conjunctival fluid is through a lick around the eye. Additional routes of infection involve wiping an eye following petting an infected cat, and other instances have been reported in which the penetrating agent was a thorn prick already licked by an infected cat. As a result, at least 25% of patients do not remember having been scratched or bitten by a cat prior to developing symptoms, despite the fact that more than 90% of patients describe encounters with new cats [8].

*Bartonella henselae* is a Gram-negative bacterium that can be included in the differential diagnosis of immunocompetent host-localized lymphadenitis. It is often acquired via scratching, biting, or close contact with cats, particularly kittens, but one instance following a dog scratch has been reported. CSD is a zoonotic illness that has no preference for race or sex. While it may affect people of any age, the overwhelming majority of documented cases involve children and teenagers. Remarkably, 90–95% of patients with CSD report having cat interactions. Nevertheless, ocular CSD has been observed in individuals without a history of cat interaction [9].

Infection with *B. henselae* is more frequent in cats in warmer climates or places with greater yearly rainfall. The occurrence peaks between October and January. Prior studies have shown that the greatest incidence of cat-scratch disease occurs in the late summer and autumn, with a second peak frequently observed in January [10,11]. Among specimens collected at Mayo Clinic Laboratories over a 10-year period, one study concluded that the prevalence of *B. henselae* seropositivity was the greatest between the months of September and January, with the highest yearly incidence in January [1]. 

In serology, seasonality seems to play a role. The explanation for the seasonal serology is due to both cat behavior and the *C. felis* life cycle [1]. Adult cat fleas feed on the blood of the host cat and spread *B. henselae*. Fleas have a four-stage life span. Humidity and temperature are key factors in flea reproduction, development, and lifespan. The seropositivity of *B. henselae* is significantly greater in cats in warm, humid areas compared with cold, dry climates, owing to the higher prevalence of *C. felis* fleas in warmer areas [10,11]. As a consequence, cats have a greater number of fleas throughout the summer and fall months than they do in the spring and winter. Sexual behavior in cats can potentially have an effect on the seasonality of CSD [10,11]. Cat reproduction is more common in spring and summer, and kittens remain with their mothers until the age of 3–4 months. Additionally, during the fall, people are more inclined to adopt kittens. *B. henselae* infection seems to be more prevalent in young cats, and the infection rates tend to fall with the duration of cat ownership [12]. Additionally, cats encounter more fleas throughout the summer and fall, which facilitates the transmission of *B. henselae* from cat to cat.

Furthermore, in [12], it was found that atypical cat-scratch illness was more prevalent in the months of August–October and January–March. The exact cause is unknown, although it may be related to delays in diagnosing atypical cat-scratch illness. For instance, individuals who acquire cat-scratch illness and experience complications in January may not receive an unusual cat-scratch disease diagnosis at that moment if they do not exhibit conventional symptoms or if their symptoms progress slowly and they do not seek treatment right away. The prevalence of *B. henselae* was found to be 8.3% in urban regions, 11.9% in semi-rural areas, and 0% in rural areas [12].

## 3. Presentation in Clinical Practice

CSD is the primary and most common clinical manifestation of *B. henselae* infection, usually manifesting as subacute localized lymphadenopathy following a cat scratch or bite [13]. Reports indicate that CSD appears more frequently in children and adolescents but less often in elderly adults and immunocompromised patients [14]. *B. henselae* infection, on the other hand, can be especially severe in immunocompromised individuals, including those with AIDS, who may develop vascular proliferative lesions [15].

Only a few investigations have been performed to examine the total mortality associated with CSD, which is usually regarded as a benign illness. Individuals with endocarditis and cerebral involvement may die due to diagnostic latency [11].

Around 95% of patients report prior cat interaction, and approximately 73% report having been scratched by a cat [16,17]. The cat flea (*Ctenocephalides felis*) was previously discovered as a disease vector arthropod [18,19]. *Ctenocephalides felis* spreads *Bartonella henselae* between cats and frequently between cats and humans through infected flea excrement, resulting in an initial inoculation lesion and lymphadenopathy resulting from the incorporation of flea excrement from *Ctenocephalides felis* in skin abrasions caused by cat scratches or bites [18,19]. Numerous *Bartonella* subspecies have been linked to human illnesses, but *Bartonella henselae* seems to be the most often involved with ophthalmic inflammations. 

Between 1.5% and 20% of patients with CSD develop nonspecific symptoms such as retinitis/neuroretinitis, conjunctivitis, neuritis, encephalitis, hepatosplenic illness, osteomyelitis, erythema nodosum, and endocarditis [20].

The most frequently observed symptom of lymphoid CSD is lymphoid CSD. The infection is often caused by a scratch or bite from a cat and is proceeded by the formation of nontender erythematous pustules or papules at the location of initial cutaneous inoculation. Individuals have flu-like systemic symptoms during the following one to two weeks, along with localized lymphadenopathy. This subtype of CSD is self-limiting and often disappears after several weeks. LAP (peripheral lymphadenopathy) is usually unilateral, involving a solitary lymph node (50%) or a group of lymph nodes (20%), sometimes many lymph-node areas (30%). LAP may be painful and suppurative in certain instances. Headaches, nausea, vomiting, anorexia, and sore throat have all been reported as symptoms (Table 2). Furthermore, some people develop a generalized maculopapular rash or erythema nodosum [21].

CSD may develop as a generalized illness in a small percentage of patients, about 5–14% [22]. The eye is the organ most frequently implicated in the widespread progression of CSD. Ocular bartonellosis occurs in around 5–10% of individuals with CSD [23]. Parinaud oculoglandular syndrome, anterior uveitis, intermediate uveitis, choroiditis, choroidal mass, retinal infiltration, neuroretinitis, branch retinal artery occlusion, serous retinal detachment, and acute endophthalmitis are all clinical manifestations of eye injury [24]. With the exception of ocular involvement, hepatosplenic illness (granuloma-tumor hepatitis, splenomegaly, or splenic abscess), pneumonia, endocarditis, encephalitis, osteomyelitis, and paronychia have all been reported [25].

Symptoms after infection are uncommon, occurring more frequently in individuals with decreased immunity. The first and most common symptom (affecting 0.3–2% of patients) is encephalopathy, which manifests in coma, convulsions, and peripheral and cortical nerve abnormalities. Additionally, it may culminate in inflammation of the retina and optic nerve, leading to unilateral visual impairment. All the symptoms should disappear within 1–3 months. Additionally, erythema nodosum, infective endocarditis, musculoskeletal pain, hepatomegaly, pneumonia, and osteomyelitis are uncommon consequences [26].

In the case of fever of unknown etiology or prolonged fever and lymphadenopathy, patients should be asked whether they have been in contact with a cat or dog, including if they have been scratched or bitten, and if they have acquired a primary skin lesion that started as a vesicle at the inoculation site. While prior exposure to cats is helpful for diagnosis, it is not required to establish a diagnosis. Additionally, in the case of manifestations such as pain, malaise, and anorexia, as well as a low-grade fever, musculoskeletal manifestations, and hepatosplenomegaly, cat-scratch disease might be suspected.

*Bartonella* PCR typically returns negative results for the blood. Regrettably, no gold standard for conclusive CSD diagnosis has been developed. Considering the technical difficulties associated with isolating *B. henselae* from patient specimens, serology seems to have become the gold standard for diagnosing CSD. This is typically accomplished using available indirect immunofluorescence assays (IFAs) capable of detecting IgM and IgG antibodies to *B. henselae* [24,25,26]. Due to its high sensitivity and specificity, real-time polymerase chain reaction (PCR) on lymph nodes or other clinical samples has been recommended as a viable approach for detecting *B. henselae* DNA in suspected cases of CSD [24,25,26]. However, this approach is restricted by the necessity for surgical collection by lymphadenectomy or biopsy, which could be addressed by conducting real-time PCR on DNA samples obtained from extracted pus or blood. Even so, this method might not be indicated in patients lacking bacterial DNAemia.

The direct identification of *B. henselae* using microbiological cultures is difficult owing to the bacterium’s slow growth rate. Although more sensitive than microbiological culture, serological analysis for anti-*B. henselae* IgM and IgG antibodies using IFA, the first-line diagnostic test for CSD, lacks specificity because of the seropositivity of many asymptomatic people due to past animal exposure [24,25,26]. Additionally, the discovery of anti-*B. henselae* IgM antibodies is a marker of acute illness that persists in the blood only for about three months after exposure. *B. henselae* IgG antibodies may be identified in the blood for up to 5–7 months after exposure, with only 25% of individuals maintaining IgG seropositivity after one year [24,25,26]. A lymph-node biopsy usually confirms it, although it is invasive. Serological testing for *B. henselae* antibodies is a commonly recognized method for the laboratory confirmation of CSD. IgM antibodies indicate acute infection. By the time a patient seeks medical care, IgM antibodies against *B. henselae* may have diminished and can occasionally be negative. In a quarter of patients, IgG titers last for a year [26]. In patients with a comparable clinical condition, elevated IgG titers are sufficient for making a diagnosis. 

## 4. Immunopathology

Every *Bartonella* species seems to be specialized to mammalian natural hosts, and *Bartonella* infection is characterized by persistent intraerythrocytic bacteremia [27]. As a consequence of intraerythrocytic parasitism, the germs may remain in the host’s circulation. The principal intracellular habitat used by *Bartonella* spp. remains unknown.

The illness manifests clinically in a variety of ways that are strongly linked to the patient’s immunological state. Cat-scratch illness often manifests as mild lymphadenopathy and fever or unspecified symptoms in immunocompetent individuals, but more severe symptoms such as hepatosplenomegaly or meningoencephalitis are seen in immunocompromised individuals [28].

No particular variables affecting the transmission of infection have yet been identified [29]. At first, the disease is asymptomatic or presents with superficial evidence of infection including a lump, lesion, or blister. By around Day 14, a severe fever and local lymphadenopathy develop. Lymph nodes become inflamed and begin to suppurate. Bacterial proliferation results in the release of proinflammatory chemicals and growth regulators and the cessation of apoptosis, which manifests as new lumps inside the vascular system. The signs can mimic cancer in half of cases, and in a minority of instances, they are connected with the musculoskeletal system, including osteitis, arthritis, and myositis [30,31]. In usually healthy subjects, the disease progresses slowly and may self-regulate. However, in several instances, antibiotic treatment is required. 

The prospect for full recovery is good in immunocompetent individuals with CSD. Considerable morbidity occurs, in 5–10% of cases, often as a result of the involvement of the central or peripheral nervous systems or as a result of multisystemic, widespread illness. A single bout of cat-scratch illness immunizes all patients for life [32].

Ninety percent of the time, CSD manifests as subacute, localized, self-limiting lymphadenitis, followed by a regional cutaneous response at the scratch site. The symptoms often resolve after 2–4 weeks. Lymphadenopathy is often self-limiting and does not need antibiotic treatment.

Around 10% of patients with CSD present with unusual symptoms such as protracted fever (longer than two weeks), erythema nodosum, and hepatic and splenic granulomas [33,34]. Skeletal involvement throughout CSD is an uncommon occurrence in the literature, presenting in 0.17–0.27% of all CSD cases.

The immune system of the infected person greatly influences the development of *B. henselae* illness. Infections from *B. henselae* are swiftly removed from the circulation and confined inside the lymphatics in people with good immune function. The immunological response can persist for 2–4 months, and this immunopathogenesis is assumed to be the fundamental mechanism implicated in the clinical condition known as CSD. Long-term systemic infections are common and may be fatal in those who are immunocompromised due to HIV or other factors. The pathogens’ intracellular survival strategies and the host’s defensive systems determine the result of intracellular bacterial infection [35]. An efficient immune response requires both humoral and cellular immunity. The host uses cell-mediated immunity to defend itself from numerous intracellular infections, but this response may cause cell death and tissue damage. Inadvertently blocking or subverting normal host cellular defense systems contributes to the etiology and illness outcome [36].

Numerous cellular mechanisms contribute to the sequence of events leading to delayed-type hypersensitivity throughout *B. henselae* infection (Table 3). CD4 Th1 lymphocytes are extremely prone to developing into effector cells and have a direct role in the control of delayed-type hypersensitivity during CSD [37]. It is well documented that macrophage mobilization and stimulation is a critical component of delayed-type hypersensitivity and is directly reliant on Th1- and NK-derived IFN-gamma induction [38]. As a component of the immune system’s innate arm, macrophage-derived cytokines serve as the first line of defense, initiating and controlling numerous inflammatory cell actions [38]. 

Auxiliary effector cells, on the other hand, are often attracted in reaction to cytokines produced by activated cells nearby. Bacterial uptake by phagocytes has been proposed as a first mechanism regulating progression and preventing *B. henselae*-induced bacteremia [39]. Indeed, a small number of in vitro investigations have implicated professional phagocytes such as macrophages as possible effector cells capable of regulating disease development or secreting angiogenic cytokines during *B. henselae* infection [40].

*B. henselae* has been shown to be intraerythrocytic in cat erythrocytes. Endothelial cells are a preferred host for *Bartonella* spp., and intracellular *B. henselae* may be detected in contaminated endothelial cells in vitro [41]. 

In their native hosts, *Bartonella* species are intimately connected with erythrocytes. While there is proof that *B. henselae* may infiltrate endothelium cells, epithelial cells, monocytes, or macrophages, there is debate regarding its potential to infect red blood cells [42]. *B. henselae* has been demonstrated to infect freshly obtained human CD34 hematopoietic progenitor cells. Humans are a reservoir for *Bartonella* spp., and the possibility of transmission through blood should be recognized. 

Human-infected *Bartonella* spp. may increase endothelial cell growth. Encouraging endothelial cell proliferation, limiting endothelial cell death, or infecting macrophages generating VEGF (vascular endothelial growth factor) induces angiogenesis [43]. Endocrine system activation, monocyte/macrophage activation, and the development and maintenance of a paracrine angiogenic loop appear to be dependent on NF-kB activation [44]. *Bartonella henselae* or *Bartonella quintana* infections may induce endocarditis and systemic vasoproliferative lesions affecting the brain, bone, bone marrow, lymph node, skeletal muscle, and mucosa [45]. Histologically, *B. henselae* infection causes endothelial proliferations with neutrophil accumulation in the interstitial space, cat-scratch lymphadenopathy, and hepatosplenic granulomas [46]. 

Bacillary angiomatosis or bacillary peliosis affects immunocompromised patients with severe HIV infection or organ-transplant recipients [47]. Granulomatous reactions are more common in immunocompetent people. Diabetes mellitus small vascular disease causes distal sensomotor polyneuropathy with autonomic nervous system involvement. It is unclear how *B. henselae* infection causes sensomotor polyneuropathy [48]. *Bartonella*’s vasculopathic activity may affect the vasa nervorum. This may be triggered directly by *Bartonella* infection or indirectly by soluble chemicals such as VEGF generated when the bacteria touch their host [49].

Endothelial cells and CD-34 hematopoietic progenitors are particularly linked to *B. henselae*. It is widely established that *Bartonella* may produce arterial infections and vasoproliferative lesions. Microorganisms of this family are quickly absorbed by endothelium cells in vitro, a mechanism mediated by actin [50]. *B. henselae* infection was already demonstrated to stimulate endothelial cell growth through paracrine vascular endothelial growth factor (VEGF) release [51]. The immunological response to *B. henselae* infection varies according to the immune status of the host. A granulomatous and suppurative response is evoked in immunocompetent people, while a vascular proliferative response is elicited in immunocompromised individuals [2].

*Bartonella* adhesin A facilitates attachment to the extracellular matrix and mammalian host cells, and another pathogenicity element is the VirB/VirD4 type IV secretion system.

The type IV secretion system VirB/VirD4 is a critical virulence factor for endothelial cell function subversion. The Trw-system was demonstrated to be the molecular determinant of host-specific erythrocyte infection [52]. A Trw-system, extra adhesins, and filamentous hemagglutinins may further contribute to *B. henselae*’s pathogenicity [7]. Immunocompromised persons (e.g., AIDS patients) who contract *B. henselae* might develop bacillary angiomatosis. The pathogen is obvious in these lesions, and its clearance by antibiotics caused the remission of angiomatous tumors [8,9]. The bacterium *B. henselae* may cause bacteremia in immunocompetent humans [10].

Adherence to the host is crucial in bacterial infection. In *B. henselae*, *Bartonella* adhesin A mediates the initial and final adhesion [11]. Trimeric autotransporter adhesins are present in several alpha-, beta-, and gamma-proteobacteria and are implicated in pathogenicity [15]. This lollipop-like surface has a modular architecture consisting of several domains [15]. The C-terminal membrane anchor is required for trimer assembly and autotransport function in all trimeric adhesins. Throughout formation, trimeric autotransporter adhesins are secreted into the periplasm. The *Bartonella* adhesin A protein is also required for the binding of extracellular matrix components to host cells, and the formation of proangiogenic host cell reactions via the modulation of hypoxia inducible factor (HIF)-1, an important transcription element involved in angiogenesis, following the secretion of angiogenic cytokines [11]. 

Nuclear factor-κB (NF-κB) promotes the expression of a variety of pro-inflammatory genes, as well as those encoding cytokines and chemokines, and is also involved in the control of the proinflammatory mediators [53]. Additionally, NF-κB is required for the viability, activation, and development of natural killer cells and inflammatory T cells. *B. henselae* promotes vasoproliferation. In fact, the vitro infection of endothelial cells with *B. henselae* causes five unique cellular changes: invasome-mediated large bacterial aggregation formation and uptake, NF-κB-dependent pro-inflammatory activation, the suppression of programmed cell death, and direct mitogenic stimulation [54]. By infecting leukocytes with *B. henselae*, leukocytes are activated and produce proangiogenic factors that stimulate endothelial cells’ growth. These actions probably contribute to vasoproliferation in lesions. With the exception of the production of *Bartonella*-containing vacuoles, these activities have been connected to the Bh VirB type IV secretion system (VirB T4SS), which is a critical virulence factor that is involved in intracellular survival and the manipulation of the host immune response to infection, and its translocated effector proteins [54].

## 5. Neurotropism

The process by which *B. henselae* expresses its neurotropism is currently unexplained; however, an immune-mediated vasculitis has been exposed as a possible causative factor [55], owing the irregular structure of the cerebral vessels, which was consistent with cerebral autoimmune arteritis in certain patients following ischemic strokes due to CSD [56].

Circulating antibodies may induce an immune response by specifically damaging the blood–brain barrier and may engage with synaptic receptors that resemble bacterial periplasmic amino-acid-binding proteins in architecture [56]. The infiltration described may well allow circulating pathogenic antibodies to enter the nervous system, allowing for interaction with brain antigens [57]. As a consequence of the resulting brain injury, convulsions could be triggered. Several clinical and experimental investigations [58,59] have revealed that focal seizures lead to a temporary increase in the blood–brain barrier’s (BBB) permeability, triggering a cascade of pathologic events that ultimately leads to the chronic activation of the epileptic focus. 

Numerous experimental and clinical studies [58,59] have demonstrated that focal seizures cause an additional transitory elevation in the BBB’s permeability, initiating a sequence of pathologic processes that finally results in the persistent activation of the epileptic focus. Given that encephalitis is an uncommon and infrequent consequence of CSD, it is reasonable to hypothesize that the condition may elicit a unique immunological response in certain people. It is possible that this mechanism might be activated against certain receptor sectors in a manner prepared as described in synapses in the pathogenesis of other cases of epileptogenic encephalitis [60], and that, based on the particular reactivity, this may play a significant role in epileptogenic induction.

Although the mechanism by which *B. henselae* demonstrates its neurotropism remains unknown, immune-mediated vasculitis has been suggested as a potential causal agent, mainly due to the irregular shape of the cerebral vessels, which displayed characteristics consistent with cerebral autoimmune arteritis in some patients who presented ischemic strokes caused by CSD [55].

*Bartonella* spp.’s capacity to penetrate a range of cell types, particularly microglia, and change surface molecules probably leads to immune evasion [61]. Therefore, *Bartonella* spp. bloodstream infections may not always result in neutrophilia, increased erythrocyte sedimentation rates, and CRP or CSF pleocytosis. Additionally, in certain cases, *B. henselae* seems to decrease immunoglobulin synthesis and natural-killer-cell activity [62].

It was discovered that *B. henselae* may invade human brain vascular pericytes in vitro [63,64,65,66]. *Bartonella* enters host cells by two means: endocytosis, which forms a *Bartonella*-containing vesicle (BCV), and invasome-mediated absorption, which results in the incorporation of huge groups of germs [64,65,66]. These findings indicate that endocytosis is the primary mode of entry for *Bartonella* into human brain vascular pericytes. Small colonies of bacteria were discovered to be connected to and absorbed by pericyte membrane infoldings. In endothelial cells, *Bartonella* exerts a clear mitogenic effect [67]. Researchers examined the impact of *B. henselae* contamination on pericyte multiplication and discovered that *B. henselae* acted as a suppressor. 

Decreased pericyte multiplication was obviously not attributable to apoptosis activation, as *Bartonella* infection had no major influence on the apoptosis of human brain vascular pericytes compared to uninfected controls. Cell death by necrosis is, indeed, an improbable process, given that the cells produced greater amounts of VEGF 72 h after contamination; however, because large MOIs were utilized and necrosis was not explicitly examined, cell damage or death cannot be ruled out molecularly. 

Due to *Bartonella henselae*’s preference for endothelial cells and their closeness to the circulation, endothelial cells seem to be a critical component of the main environment. The endothelial space enables the bacteria to sporadically populate the circulation with organisms, possibly infecting CD34+ precursor cells in the bone marrow and also circulatory erythrocytes and monocytes [68]. This could be a secondary explanation for the results of the brain MRI scans. Nevertheless, the process by which *Bartonella* induces CNS infection is unknown.

Angiogenesis starts with the detachment of pericytes from endothelial cells, which promotes endothelial cellular growth, resulting in neovascularization [69]. The number of pericytes surrounding the parental arteries decreases during this phase [69]. The newly established vasculature is maintained by pericyte induction [70]. Although the process by which *B. henselae* inhibits pericyte growth is unknown, in the context of angiogenesis, inhibiting pericyte development in freshly emerging arteries could result in decreased pericyte coverage and increased vascular growth.

VEGF and angiopoietin-2 (Ang-2) are critical variables in the separation of endothelial cells and pericytes [71,72]. Pericytes generate VEGF in hypoxic conditions, enabling their separation from endothelial cells and serving as a paracrine endothelial cell mitogen [72]. VEGF may also function as a stimulator for pericytes’ multiplication in hypoxic settings [73]. The inoculation of human brain vascular pericytes with *B. henselae* increased VEGF in a dose-dependent manner [74]. The involvement of VEGF in *Bartonella*-induced vasoproliferation is well established [74]. Vascular endothelial cells generate insufficient VEGF in relation to *Bartonella* infection. *Bartonella* infectious disease stimulates the NF-kB pathway [46], resulting in the recruitment of inflammatory cells, including macrophages, while macrophage colonization with *Bartonella* results in the stimulation of hypoxia inducible factor-I and the secretion of VEGF, which can then act in a paracrine fashion on endothelial cells, promoting their expansion [74]. 

Infection with *B. bacilliformis* stimulates the synthesis of Ang-2 by endothelial cells [47]. Ang-2 competes for binding to Tie-2 receptors with Ang-1 (which is crucial for creating pericyte–endothelial cell contacts, resulting in the suppression of endothelial development and vascular stability). *Bartonella* spp. may promote both VEGF and Ang-2 production [47,74], and their combined effect may result in decreased pericyte coverage, encouraging angiogenesis in a manner analogous to oxygen-deprivation tumor angiogenesis. IL-8 is also a cytokine that stimulates angiogenesis, which promotes vasoproliferation in response to *Bartonella* infection [43]. IL-8 stimulates angiogenesis by increasing endothelial cell survival and proliferation, and the synthesis of matrix metalloproteinases, which aid in the initiation of angiogenesis through the breakdown of the basement membrane [75]. 

In the work of Varanat et al. [76], *Bartonella* infection had no impact on pericyte-derived IL-8 secretion. Gram-negative bacteria’s lipopolysaccharides are a recognized inducer of IL-8 [77]. *Bartonella* spp. have been shown to increase the production of IL-8 throughout the human bloodstream [78], macrophages, the hepatocellular area, and endothelial cells [79], and IL-8 has been shown to enhance angiogenesis by preventing endothelial cell death. The findings of Varanat et al. [76] suggest that pericytes are unlikely to be a major producer of IL-8 during *Bartonella*-induced vasoproliferation. There may be a link between the production of IL-12 and the avoidance of pathologic angiogenesis in immunocompetent patients in *B. henselae* infection [80]. Given that *B. henselae* largely elicits a Th1 response, future research should examine the involvement of IL-23 and IL-27 in the immune reaction to *B. henselae*.

Munana et al. [63] found that *B. henselae* could invade and persist intracellularly, lasting up to 30 days after implantation without causing host cell ultrastructural injury, indicating that chronic infection of the central nervous system could occur. In certain parts, a bilayer membrane around the bacteria was observed, as were vacuoles inside a few of the organisms. Infected cells exhibited no morphologic alterations, unlike those in noninoculated conditions. Such findings could contribute to our knowledge of the neuropathogenesis of *B. henselae* infection and indicate the possibility of chronic CNS infection with *B. henselae*. Infection with *B. henselae* has been associated with neurological impairment in humans, most often in individuals with cat-scratch illness [63].

Manana et al.’s findings on felines demonstrate that *B. henselae* preferentially affects the glial central nervous system. Microglia are produced from stem cells of the bone’s precursor cells and function as the brain system’s resident tissue macrophages. Prior studies have established that macrophages might also be infected by *B. henselae*; the pathogen was shown to invade murine macrophage-like cell cultures [81], and the Warthin–Starry silver staining of lymph-node samples from humans with cat-scratch disease has distinguished a variety of organisms inside tissue macrophages [82]. 

Additionally, research employing a mouse model of the immune response to *B. henselae* infection reveals that macrophages play a pathogenic role in infection. The spread of the organism through macrophages would allow it to penetrate the CNS, since peripheral blood phagocytes are able to traverse the blood–brain barrier. Once within the brain, the infection of local microglia can aid in the establishment of chronic infections.

The infection of microglial cells can occur as a consequence of the organism’s trophism towards microglial cells or as a consequence of host-cell-mediated phagocytosis. The in vitro infiltration of endothelial cells by *B. henselae* was demonstrated [50]. 

Given the considerable number of publications indicating CNS illness caused by *B. henselae*, no evidence of the infection of the brain parenchyma in immunocompetent human or feline hosts has been established. The fleeting and severe character of the clinical symptoms in people with cat-scratch-disease-related encephalopathy has prompted researchers to speculate about an alternative pathophysiologic process. *B. henselae* might elicit an immunological response or result in the release of a neurotoxin [83].

The absence of any detectable cytotoxic action in infected cells indicates an intermediate disease process. Microglia are a plausible target for these processes. The cells are critical for maintaining homeostasis inside the CNS and for the local modulation of immunological and inflammatory responses. Interleukin-1 (IL-1) and tumor necrosis factor alpha (TNF-α) are inflammatory cytokines produced by activated microglia [84].

The production of IL-1 and TNF-α by microglia has been demonstrated to recruit immune cytokines into the CNS and to encourage the proliferation and production of other soluble factors by endothelial cells and astrocytes [85]. Increased local cytokine generation may be detrimental for CNS cellular components, as evidenced by TNF-α’s effects on myelin and myelin-forming cells [86]. Together, IL-1 and TNF-α produce fever, promote inflammation and acute-phase effects, and are cytotoxic to some brain cell lines in vitro, which may even suggest the induction of encephalopathy [84].

Additionally, infection could affect the normal structural relationships among brain cells in vivo. Microglia generate significant branching of cellular functions surrounding neurons, and their physical closeness implies that this relationship has a functional importance [85]. By altering the usual microglial–neuronal connections, changes in contaminated microglia could have severe functional repercussions on neurons. 

## 6. Neurologic Manifestations

Neurologic complications from *B. henselae* infection are infrequent, occurring in around 2–7% of infected persons. Neurological symptoms often manifest two weeks after the onset of fever and lymphadenopathy [87,88]. Recently, *Bartonella* infections have been related to a broader range of neurological symptoms, including hallucinations, loss of weight, muscular exhaustion, partial paralysis, and pediatric acute-onset neuropsychiatric syndrome (PANS) [89].

The findings imply that this species of bacteria may be capable of causing a spectral range of neurological symptoms, varied clinical features, and multi-organ system pathology that differs in intensity and screening difficulty across an extended time period, varying from weeks to years [52,54]. 

The majority of individuals with CSD who have central nervous system damage are children under the age of 18. One of the most common symptoms is encephalitis, although radiculitis, polyneuritis, myelitis, and cranial nerve involvement have all been described [90,91]. Neurologic alterations often occur about two weeks after the start of fever and significant lymphadenopathy, which are frequent in these patients. Seizures are the most frequent first symptom, but headaches, alterations in mental state, delirium, and coma are all possible [91]. Most individuals recover completely without complications in under a year [92]. Computed tomographic imaging revealed no abnormalities or localized low-density lesions; an electroencephalogram demonstrated generic, widespread slowness [91]. The pathologic characteristics of CSD in the brain are unknown, as these tissues are rarely biopsied. 

However, granulomatous inflammation of the brain tissue involving meningitis has been demonstrated, and special stains and molecular testing have shown that organisms recover from such lesions [93]. Usually, the diagnosis is established based on the findings of a lymph-node biopsy or other supportive laboratory tests.

Except for the detection of *B. henselae* infection, the laboratory examination of infected individuals with encephalopathy usually produces inconsistent findings and is thus ineffective for diagnosis.

Electroencephalography conducted during the acute period of illness shows a widespread slowness in about 80% of patients, which resolves completely upon follow-up [33]. Approximately 19% of patients show abnormal results on a CT scan or magnetic resonance imaging (MRI) of the brain, which would include anomalies in the brain’s white matter, striatum, thalamus, and gray matter [94]. The outcome for individuals with encephalopathy is usually good, with 90% of patients recovering completely and spontaneously with no complications [88]. Just one case of fatal meningitis and encephalitis in an immunocompetent infant has been reported in the scientific literature as a consequence of *B. henselae* disease [95].

Examination of the cerebrospinal fluid may reveal moderate pleocytosis and increased protein content [91] but, frequently, CSF study findings are benign or suggest relatively moderate inflammation. Examination of the cerebrospinal fluid may reveal moderate pleocytosis and an increased protein content [91]. The human illness has some features in common with experimental cat-brain disease, including the temporary nature of the disruption, the existence of behavioral changes, a decline in alertness or convulsions, and the absence of major CSF irregularities. It is possible that *B. henselae* infects both species through comparable pathophysiologic processes.

With the development of increasingly accurate and precise screening procedures, recent microbiological studies have confirmed blood and cerebrospinal fluid (CSF) infections with one or more *Bartonella* species in patients with neurological, neuropsychological, and psychiatric manifestations [89]. Pediatric acute-onset neuropsychiatric syndrome (PANS) is a medical illness in children and adolescents marked by the abrupt development of neuropsychiatric symptoms such as obsessions or compulsions or dietary restriction [96]. Depression, anger, anxiousness, and academic deterioration are frequently associated with PANS manifestations. PANS may be induced by infection, electrolyte imbalances, and other inflammatory responses [96]. After failing to react to psychiatric combination therapy and an immunomodulatory operator for suspected immune-driven encephalitis, Breitschwerdt and colleagues reported a case of *Bartonella henselae* bacteremia in a child with schizophrenia who had symptom remission after antibiotic treatment [89].

Furthermore, individuals with bartonellosis with neurological symptoms may acquire autoantibodies, lacking normal clinical, pathological, and immunologic markers of bacteremia, and may be immunocompromised as a result of medication therapy or lengthy *Bartonella* spp. infection. When autoantibodies are documented in combination with negative indicators for inflammation, the clinical presumption of an infectious etiology in individuals with neuropsychiatric symptoms is reduced [61].

### 6.1. Encephalopathy

The most typical feature is encephalopathy, which presents in 90% of cases involving the neurological system. The neurological symptoms often manifest themselves two to three weeks following the start of lymphadenopathy. Headaches and an altered mental state are common symptoms. Seizures occur in 46–80% of individuals infected with *Bartonella* encephalopathy, with some individuals diagnosed with status epilepticus [88,97]. 

However, as many as 40% of individuals with *Bartonella* encephalopathy have been observed to exhibit aggressive behavior [98]. Along with an altered mental state, individuals with encephalopathy can exhibit a range of neurologic findings, such as weakness, muscle-tone changes, nuchal stiffness, extensor plantar reflexes, and hypo- or hyperreflexia [83].

Cat-scratch disease encephalopathy exhibits several features, including a prolonged time interval between infection and the progression of CNS symptoms, an accelerated initiation of significant brain impairment, and a potential and, at times, extremely rapid resolution of symptoms without residual neurologic deficits. Compartmental disorders, mental decline, and convulsions are all frequent signs of illness [33]. 

The link between CSD and encephalopathy remains largely unknown. Encephalopathy was initially defined as having neurological involvement in CSD 70 years ago [98] and is often associated with a favorable long-term result [33]. Numerous instances of encephalopathy have been documented in recent times. Such encephalopathies may involve the brainstem [99] or striatum [100], and may result in infarction because of a vasculitic complication [101] or status epilepticus in children [102]. Numerous hypotheses regarding the etiology of CSD encephalopathy exist, including direct penetration, neurotoxins, and vasculitis due to an immunological response [103].

### 6.2. Encephalitis

Encephalitis was the most frequently encountered neurologic symptom in the population studied by Nawrocki et al., which was in line with the observations of Reynolds et al., who found that the majority of children hospitalized for the neurologic complications of cat-scratch disease were diagnosed with encephalitis or encephalopathy [93]. Hence, doctors must investigate cat-scratch disease in individuals with encephalitis or newly diagnosed hepatosplenic disorders, particularly in children.

Considering the nature and infrequency of encephalitis as a result of CSD, it is fair to postulate that the disease might evoke a distinct immune response in some individuals. It is conceivable that this process could be enabled against specific receptor sectors in a way outlined in synapses in the pathogenesis of other epileptogenic encephalitis, and that, depending on the specific reactivity, this could contribute to the formation of the epileptogenic process [60].

When evaluating all the etiologies of juvenile viral encephalitis, focal neurologic symptoms are prevalent [104]. In a massive study by Kolski et al. [104], around 60% of juvenile encephalitis cases were preceded by partial epilepsy, and 50%, by other focused neurologic symptoms. Encephalopathy, convulsions, status epilepticus, retinitis, transverse myelitis, and cerebral vasculitis are among the presentations recorded in individuals with cat-scratch illness who have neurological signs [105,106,107]. Encephalopathy has been the most prevalent neurologic symptom of cat-scratch illness and may occur many weeks after the first contact with a cat [108]. 

Cathiers et al. [33] examined the prevalence of bartonellosis’ various neurological symptoms, with encephalitis being the most common. The age range of the encephalopathy patients was 10.6 years. Convulsions happened in 46%, whereas confrontational behavior occurred in 40%. Without neurologic sequelae, all the patients (*n* = 76) recovered within 12 months, and 78% in 1–12 weeks. However, there have been reports of neurological symptoms persisting for more than a year [101].

According to Fouch et al., a six-year-old boy arrived with a lymphatic axillar nodule, and his condition deteriorated over the following few days, with headaches, seizures, and mental state abnormalities, which culminated in a vegetative state and the patient’s death. A postmortem examination revealed widespread and severe perivascular lymphocytic infiltrates with microglial nodules across the frontal, parietal, and occipital lobes, as well as the pons. *Bartonella henselae* was identified as the causal agent [109]. 

A recent study has shown a variety of antineuronal autoantibodies in the circulation and CSF of certain patients with status epilepticus and encephalopathy of unclear cause, indicating that autoimmunity may play a role in these instances [110]. These autoantibodies may be oriented against a range of different cell-surface antigens, and several have been linked to particular seizure conditions, including limbic encephalitis. If autoantibodies are detected in cat-scratch disease encephalopathy, immunotherapy in the acute stage, such as injectable immunoglobulins or systemic steroids, may be tried. 

### 6.3. Other Neurological Manifestations

Less-frequent neurologic consequences include meningomyeloradiculopathy, which manifests as paresthesias, paralysis, and sphincter dysfunction in the lower extremities, facial nerve palsy, Guillain–Barré Syndrome [110,111], continuous partial epilepsy [112,113], acute hemiplegia, transverse myelitis [114], and cerebral arteritis [56]. In research by Easley et al., two children were reported who first presented with status epilepticus and normal CSF fluid tests [115].

Multiple-sclerosis-related demyelinating illness [116], acute stroke-related CNS disease [103], transverse myelitis [117], psychiatric conditions [118], and persistent intellectual and locomotor impairments [89] are further types of central nervous system dysfunction. 

Facial nerve invasion is quite uncommon in *Bartonella* infection. Walter et al. [110] described the first instance of acute facial nerve paralysis in a cat-scratch disease patient with characteristic lymphadenitis who was immunologically confirmed. 

According to Nakamura et al. [119], a seven-year-old child presented with fever, cervical lymphadenopathy, and peripheral facial nerve palsy, all of which were serologically confirmed to be due to cat-scratch illness. On the left, the stapedius muscle reflex was missing, and magnetic resonance imaging of the brain showed a mass lesion in the left internal auditory meatus. Mutucumarana et al. [120] described a five-year-old girl who had a nearly three-week-long fever, along with exhaustion, headaches, a six-pound weight loss, and late-onset left-sided facial palsy. 

Premachandra et al. [121] documented a case of a child who had a parotid lump and a lower motor neurone facial paralysis affecting the marginal mandibular branch of the facial nerve, suggesting a malignant parotid tumor. Additionally, Ganesan et al. [111] described a novel etiology of parotid edema, brief facial paralysis, and mild ptosis. Chiu et al. [122] documented the case of a 10-year-old child who had two weeks of left facial paralysis. The patient presented with left-eye conjunctivitis, facial edema, and weakness, and also had a severe left neck mass. Unilateral facial nerve branch paralysis was discovered during the test. Thompson et al. [123] documented an adult with peripheral facial nerve palsy caused by cat-scratch illness; the patient also had neuroretinitis.

In the report of Rocha et al. [114], a child presented with left hemiplegia and a right frontoparietal lesion on head neuroimaging due to cat-scratch illness. Another report by Cerpa et al. [124] was regarding a child who had transitory right hemiparesis during cat-scratch disease with status epilepticus, a finding consistent with Todd’s paralysis. Furthermore, two cases of cat-scratch-disease-associated vasculitis as the cause of stroke and subsequent hemiparesis have been reported [125]. Vermeulen et al. performed a retrospective analysis and identified 20 instances of cat-scratch disease-related spinal osteomyelitis, one of which had severe transitory paresis [126]. From these findings, it can be concluded that the flaccid paralysis associated with cat-scratch-illness encephalitis may be triggered by a specific contamination of the right hemisphere, postictal Todd’s paralysis, or spinal-cord invasion. 

Yep et al. [127] reported on the case of a young woman presenting with meningitis and neuroretinitis who had fever, sweating, a left frontal headache, hazy vision, and painful left-eye movements for five days. A central scotoma decreased color vision, and enlargement of the optic disc and macula were seen in the left eye. The patient reported cat contact a few months prior to the beginning of the symptomatology. A thorough examination of the fundus with a tropicamide-dilated pupil revealed a severely enlarged macula in addition to optic nerve damage.

Anabu et al. described a seven-year-old boy who acquired choreoathetosis after CSD encephalopathy and showed abnormal basal ganglion alterations on an MRI scan. He also developed convulsions with a fluctuating level of consciousness; initially he was still unable to walk or talk, though after several weeks, his condition improved—but without a complete recovery [100].

## 7. Ocular Manifestations

*B. henselae* was discovered to be the causal agent of this disease in only the past decade [128]. The infection is believed to be spread through direct conjunctival inoculation. Common symptoms include a foreign-body sensation, unilateral eye redness, serous discharge, and increased tear secretion. Patients usually present necrotic granuloma with conjunctival epithelium ulceration, and also local lymphadenopathy involving the preauricular, submandibular, or cervical lymphatic system [129]. After many weeks, the granuloma often resolves without scarring.

Ocular involvement develops in 5–10% of people with CSD [33]. The eye could serve as the major infection site, resulting in Parinaud oculoglandular syndrome, which is defined by the infection of the conjunctiva and eyelids, as well as local lymphadenopathy (Figure 1). Within 2–3 weeks of the onset of systemic illness, various ocular symptoms might arise. Neuroretinitis, optic neuropathy, and a variety of other kinds of intraocular inflammation are among these symptoms. Regarding the annexal manifestation, Parinaud Syndrome would be representative. Considering vitreous changes, intermediate uveitis and vitreous hemorrhage can be present. Inner retinitis and chorioretinitis can also be included in the retinal and/or choroidal manifestations. The retinal vascular manifestations are retinal vasculitis, angiomatous-like proliferation, retinal arteriolar branch occlusion and retinal vein branch occlusion. The macular complications include serous macular detachment, macular stars (stellar maculopathy), macular edema, and macular hole. The optic-nerve manifestations are considered to be neuroretinitis, optic-disc edema, and optic-nerve head mass.

The most frequent clinical observations in previous investigations have been neuroretinitis associated with macular stars [24] and/or isolated foci of retinitis or choroiditis [130]. Macular stars can become visible in a few days as vision loss begins and become increasingly apparent over the following two to three weeks. Isolated ocular neuritis was also reported before [131], highlighting the necessity of eliminating infectious agents such as *Bartonella henselae* prior to initiating therapy with pulse methylprednisolone, particularly in children.

The authors of a Turkish study of 13 eyes of 10 patients with ocular bartonellosis highlighted the variety of ocular symptoms, including an acute endophthalmitis case [131]. Nine patients had a history of cat interaction and systemic symptoms for up to three months before presenting with ocular symptoms, and three had them earlier, but they had initially been misinterpreted as noninfectious intraocular inflammation [131]. The lack of vitreous cells helped to distinguish these lesions from the retinitis/retinal infiltrates observed in Behcet disease or toxoplasmosis, the investigators reported. They advised the careful monitoring of retinal infiltrates simulating cotton-wool patches, since these may progress to a branch retinal artery occlusion [131].

Ocular CSD may potentially manifest as occlusion of the retinal artery, retinal infiltrates resembling cotton-wool exudates, or endophthalmitis [132]. The pathogenesis of retinal infiltrates is most likely related to ischemia caused by the occlusion of the retinal arterioles [19]. It is essential for the ophthalmologist to distinguish the superficial retinal infiltrates in ocular CSD from retinitis or retinal infiltrates seen in individuals with sarcoidosis, Adamantiades-Behcet’s disease, toxoplasmosis or rickettsia infection [131].

Previously, Eiger-Moscovich et al. [133] described the clinical course of BRAO (Branch Retinal Artery Occlusion) in six individuals (seven eyes). One suffered numerous arterial occlusions. BRAO was found in three eyes. In one patient with unilateral BRAO, the other eye had neuroretinitis and, in two cases, both appeared in the ipsilateral eye. BRAO developed focal retinal infiltrates in both eyes or in one eye. A central scotoma left just one eye with low visual acuity (1/60). However, BRAO caused persistent visual field reduction in both eyes [133]. The authors recommended careful history-taking and serologic testing for *Bartonella* infection, particularly in children with BRAO.

Numerous case reports and series on ophthalmic vascular occlusions as a result of CSD have been published [133,134,135,136]. Surprisingly, the changes and distortions in visual acuity are location-dependent. In some instances of secondary epiretinal membranes, the epiretinal membrane (ERM) spontaneously releases. The therapy, it has been claimed, may promote this occurrence [137,138,139]. The release of the ERM (epiretinal membrane) results in a reduction or elimination of the macula’s tractional pressures, thus allowing the macular hole to close.

When a persistent tractional epiretinal membrane develops (as a consequence of uveitis), even without the formation of a macular hole, pars plana vitrectomy and ERM peeling are the recommended treatment options. In immunocompromised or immunosuppressed individuals, CSD is anticipated to present a more severe systemic effect. It has been observed that, in HIV-positive individuals, it causes bacillary angiomatosis [140].

Best et al. [141] reported a case in which a patient came to the emergency department with a unilateral central scotoma and no prodromal symptoms, a rare presentation for this illness. The patient was afebrile and had no visible skin lesions associated with a cat scratch or bite, nor did she have lymphadenopathy. An ophthalmologic examination revealed findings suggestive of retinal edema.

Suhler et al. [142] discovered that *B. henselae* infection was the most frequently observed etiology of neuroretinitis in their clinic in the northwestern USA. The degree to which these results apply to certain other areas is unclear.

Tey et al. [143] found that 10 out of 13 patients (76.9%) exhibited posterior segment symptoms, whereas three (23.1%) had Parinaud’s oculoglandular syndrome, lacking the participation of the posterior segment. Eighty-two percent of the eyes had small foci of retinal white lesions. Subretinal fluid (SRF) beneath the fovea was visible on SD-OCT in eight eyes (34.8%), and Parinaud’s oculoglandular syndrome, in four eyes (17.4%).

In the case described by Annoura et al. [144], Parinaud oculoglandular syndrome, anterior uveitis, neuroretinitis, a serous retinal detachment, and a flame-like retinal hemorrhage suggesting retinal vein blockage were all identified. Additionally, the frequency of concurrent eye disorders in the anterior and posterior chambers is notable. Additionally, the present instance was noteworthy in that ocular involvement occurred in the absence of systemic diseases, and neither lymphatic nor systemic disease emerged.

### Neuroretinitis

Neuroretinitis is characterized by unilateral vision loss, optic-disc edema, and macular star formation. Given the appearance of the macular star, it was first believed to be a macular disease and was labeled as such [145]. Soon afterwards, according to Gass et al., it was revealed that the macular star was caused by leakage resulting from the optic-disc edema, not by a retinal process in itself, so the optic-nerve edema preceded the macular star [146]. The causes of neuroretinitis are diverse, and the typical components of the triad do not usually manifest simultaneously, especially in the early stages of the illness.

Neuroretinitis is described as inflammation of the optic nerve and peripapillary retina, marked by edema of the optic disc and the eventual development of a macular star. Neuroretinitis is often unilateral and might be bilateral in immunocompetent as well as immunocompromised people [147]. Ocular symptoms often manifest 2–3 weeks following the onset of systemic symptoms. The most prevalent ocular symptom is eyesight loss [148]. A subjective afferent pupillary deficiency, a visual field defect, and dyschromatopsia are often present [148]. On occasion, cells and flares are visible in the anterior chamber, and moderate vitritis is prevalent. The fundoscopic observations often include edema of the optic disc and lipid exudation in the macula in the form of a whole or partial star [149]. A half star shape is most often found in the nasal macula.

In neuroretinitis, the optic disc is the major site of inflammation [146]. At first glance, the macroscopic star could be missing. It typically resolves 1–12 weeks following the beginning of optic-disc edema. The disc edema starts to subside after two weeks and often resolves completely after 8–12 weeks. The key role of nicotinamide adenine dinucleotide (NAD+) in retinal cells’ metabolism might influence the recovery potential of the retina, as already revealed by research regarding degenerative retinal conditions [150]. The macular star disappears after four weeks but could remain for almost a year [151]. The extreme sensitivity of macular cells has to be considered as an important factor in the recovery of the macula [152,153].

Neuroretinitis usually follows a feverish illness with lymphadenopathy, rashes, arthralgia, and headache. The ophthalmologic findings include reduced visual acuity, dyschromatopsia, a relative afferent pupillary deficit, and visual field abnormalities including a cecocentral/central scotoma. However, 2–6 weeks after the onset of symptoms, stellate macula exudates (the ‘macular star’) form. Due to the slow development of this finding, the maculopathy is frequently missed at first presentation [151]. Weeks later, individuals may develop multifocal deep-white retinal lesions and vitreous irritation. Anterior chamber response is uncommon [154]. The macular star may last for months and impede the recovery of visual acuity.

In two previous studies, retinal or choroidal white lesions predominated. According to Solley et al. [130], they were found in the superficial retina (30%), deep retina (49%), full-thickness retina (14%), and choroid (7%). Neovessels were discovered during an OCT angiography of a *Bartonella* focal chorioretinitis. It may assist in differentiating *Bartonella* lesions from other noninfectious or infectious fundus lesions because *Bartonella* spp. have a unique vascular proliferative characteristic [50].

In neuroretinitis, optic-disc edema causes serous retinal detachment, resulting in immediate unilateral vision loss, followed by macular exudates organized in a partial or full star pattern around the fovea [155,156]. OCT showed subretinal fluid in all eyes with *Bartonella* neuroretinitis, while ophthalmoscopy or fluorescein angiography did not. According to Parikh et al. [157], OCT may be helpful in detecting intraretinal edema associated with optic-disc inflammation preceding serous retinal detachment and subsequent macular-star formation. In 53 patients (62 eyes) with *Bartonella* optic neuropathy, the authors found that the frequency of optic neuropathy surpassed that of neuroretinitis, and that the lack of a macular star did not rule out optic neuropathy [158]. The majority of patients had unilateral involvement, 58% had prodromal systemic symptoms, and 26% had a history of cat encounters (53%). An exposure history was not required to diagnose ocular bartonellosis, even in areas abounding with cats.

Optic-nerve involvement such as a peripapillary angioma or disc granuloma is uncommon. Ocular granuloma with pineal-gland enlargement, indicating pineoblastoma, has recently been described by Aziz et al. [159]. In this case, enucleation of the globe revealed suppurative granulomatous inflammation surrounded by vascular growth, indicative of *Bartonella* granuloma [159]. The authors summarized 15 reported instances of optic-disc granulomas associated with CSD [159]. Moreover, Freitas-Neto et al. [160] showed that multimodal fundus imaging assisted in diagnosing CSD in a patient with a peripapillary angiomatous lesion.

## 8. Therapy

Fundus imaging, optical coherence tomography (OCT), and fluorescence angiography can aid in the diagnosis and treatment of CSD [161]. Late leakage from the optic disc is the most frequently observed finding in FA. Peripapillary angiomatosis and peripapillary serous retinal detachment are additional FA observations [129,162]. Although neuroretinitis is a well-described complication of *B. henselae* infection, angiomatous lesions remain uncommon. 

Habot-Wilner et al. [163,164] have shown the use of retinal OCT in evaluating macular alterations in CSD. The research discovered a lowering of the foveal contour, hypertrophy of the neurosensory retina, and production of subretinal fluid in each of the tested eyes. In the outer plexiform layer, the retinal exudates looked hyperreflective. Follow-up examinations revealed that the macula was normal. The authors also recommended the use of OCT in conjunction with other imaging modalities to monitor patients with CSD neuroretinitis [163,164].

*B. henselae*-induced ocular inflammation is often benign and self-limiting, with good visual outcomes. However, significant, irreversible visual loss is possible. The indication for therapy varies according to the severity of the condition (Table 4). In vitro, *Bartonella henselae* is sensitive to a variety of antibiotics [165]. There is no conclusive randomized clinical research demonstrating the effectiveness of antibiotic or corticosteroid therapy for *B. henselae* infection, but an uncontrolled study indicated the effectiveness of antibiotic therapy [165].

While *B. henselae* seems to be the most frequently observed infectious cause of neuroretinitis, treatment persists due to the disease’s self-limiting nature. Thus, according to two specialists, the rarity of *Bartonella* neuroretinitis complicates treatment [166]. Lee et al. believe that antimicrobial treatment reduces the duration of the disease symptoms and that, therefore, it should be treated with antibiotics until the laboratory testing is completed [167]. 

On the other hand, Bhatti asserts that therapy has no effect on the cure rate [166]. Given the rarity of *Bartonella* neuroretinitis, recruiting participants for a randomized clinical study would be very challenging and would require multicenter research [168]. Reed et al. [48] found that antibiotics resulted in a faster visual recovery and a shorter duration of illness. Chiu et al. [122] found no difference in visual results between treated and untreated patients (antibiotics, CS, or both). Rostad et al. cured an eight-year-old child of *B. henselae* with oral doxycycline (60 mg) every 12 h [169].

Chiu et al. found no link between systemic corticosteroid usage and visual results [122]. Meanwhile, Habot-Wilner et al. [163,164] found, in a multicenter retrospective cohort analysis, that a combination antibacterial and systemic corticosteroid regimen was related to a better visual result than an antibiotic-only regimen in patients with BCVA worse than 6/9 at presentation. Kodoma et al. described a case study of 14 patients in Japan, 13 of whom received systemic steroids and two of whom received 1000 mg of methylprednisolone pulse treatment [170].

In Garcia’s case report, oral doxycycline and rifampicin were effectively used to treat an elderly patient with a parotid abscess and aseptic meningitis [13]. The majority of the complications resolve spontaneously, while recovery from central nervous system symptoms might take up to a year [4]. There is insufficient evidence to support the advantages of particular antibiotic treatment in immunocompetent individuals with uncommon CSD manifestations [171]. Numerous antibiotic treatments, particularly gentamicin, trimethoprim/sulfamethoxazole, azithromycin, ciprofloxacin, erythromycin, doxycycline, and rifampin, were utilized in atypical CSD, alone or in various combinations.

Bejarano et al. reported a case in which a child presenting fever and tonic–clonic seizures was effectively treated with an approximately four-month course of azithromycin and rifampin [172].

Taking into consideration the available research, status epilepticus caused by this illness is often resistant to anticonvulsant medication. Even with the prompt initiation of treatment and the usage of various anticonvulsants, seizure management might not be achieved for several hours [173]. 

Due to their deep penetration of the central nervous system, the majority of researchers and doctors strongly advise a protocol of doxycycline and rifampin for 10–14 days [102], with a treatment course of 2–4 weeks in immunocompetent individuals and four months in immunosuppressed patients, but the antibiotic treatments used may vary in terms of the drug category and length of time. Antiepileptics usually cure the majority of patients within three days to two years [172]. Bogue et al. revealed that clinical signs of systemic CSD rapidly resolved in three critically sick patients following intravenous gentamicin-sulfate therapy [174].

The course of therapy would be determined by the responsiveness and the awareness that, in the normal course of the illness, recuperation from CSDE is typically quick (4–14 days). Collipp [175] noted an outstanding recovery due to trimethoprim–sulfamethoxazole in 11 CSD cases. Along with seizure management and respiratory assistance, the most critical part of treating individuals with encephalopathy is establishing a diagnosis as soon as possible. This may minimize lengthy and unsatisfying investigations that may cause the patient distress. During a coma, supporting measures such as adequate nursing care and constant monitoring of the patient are suggested. 

Balakrishnan et al. [101] reported on an 11-year-old girl who suffered from a variety of neurological symptoms, which included migraines, multisensory hallucinations, nervousness, visual impairment affecting both eyes, episodes of generalized paralysis, VII-th nerve palsy, severe sleeping problems, convulsions, extreme fatigue, cognitive deficits, and memory problems. Physiopathologically, originally, *B. henselae* generated vasculitis, which resulted in subsequent cerebral ischemia, tissue destruction, and surgical excision.

Rosas et al. [176] described a case of cat-scratch-illness encephalitis associated with left-arm flaccid paralysis. The patient manifested an altered mental state and a probable seizure during the acute phase. Initially, hemiparesis was thought to be a sign of Todd’s paralysis. Clinically, the patient recovered on the fifth day of hospitalization, regaining use of her left arm, and recovered completely a week afterwards with doxycycline and levetiracetam treatment.

Raihan et al. reported four instances in which patients had a background of interaction with cats, had a fever prior to developing ocular symptoms, had optic-disc enlargement and macular edema, and tested positive for *B. henselae* [177]. 

Zakhour et al. [178] described a 10-year-old girl who had transverse myelitis and Guillain–Barré syndrome as a consequence of cat-scratch illness. Lower-extremity weakness and sensory loss, as well as lower-extremity discomfort, were all associated with the symptomatology. The treatment consisted of doxycycline for 14 days and five days of rifampin and intravenous immunoglobulin, and the case evolution was positive; at the end of the treatment, the patient only presented mild sensory deficits. 

In the study conducted by Bilavsky et al. [179], over a period of 11 years, of cat-scratch disease in eight pregnant women, it was found that five of the eight had classic cat-scratch illness, including regional lymphadenitis; two experienced local lymphadenitis accompanied by arthralgia, myalgia, and erythema nodosum; and one had neuroretinitis. 

**Table 4 brainsci-12-00217-t004:** Treatment of cat-scratch disease with neuro-ophthalmologic features.

Author	Site of Lesion	Treatment		Dosage	Time
		Antibiotics	Corticoids		
Lee et al. [167]	Nodule of the upper lid	Topical gentamicin and systemic erythromycin	-	N/A	N/A
Kodoma et al. [170]	Neuroretinitis	14 patients: Antibiotics: ciprofloxacin, doxycycline, sulfamethoxazole, erythromycin or cephems	14 patients: prednisolone, betamethasone, methylprednisolone	N/A	N/A
Garcia Garcia et al. [13]	Parotid gland abscess and aseptic meningitis	Doxycycline and rifampicin	-	N/A	Two weeks
Canneti et al. [171]	Thirty-nine CSD patients	31 patients (8 patients with neurologic manifestations of CSD)	2 patients with neurologic manifestations of CSD	N/A	N/A
Bejarano et al. [172]	encephalopathy	Clarithromycin (5 days), cefotaxime (3 days), Meropenem (2 days), Vancomycin (2 days), Piperacillin-tazobactam (5 days), Azithromycin (134 days), rifampin (134 days)	-	Clarithromycin 15 mg/kg/day, cefotaxime 90 mg/kg/4 h, meropenem 40 mg/kg/8 h, vancomycin 10 mg/kg/6 h, piperacillin-tazobactam 80 mg–10 mg/kg/6 h, azithromycin 10 mg/kg/day	NA
Armengol et al. [97]	encephalopathy	Erythromycin	-	NA	5 days
Fouch et al. [109]	encephalitis	Cephalexin	-	NA	7 days
Balakrishnan et al. [101]	Vasculitis, cerebral infarction	Azithromycin Ceftriaxone		Azithromycin 500 mg Ceftriaxone 2 g	Azithromycin 8 weeks Ceftriaxone 8 weeks
Cerpa et al. [124]	Encephalitis with convulsive status	Ciprofloxacin Cotrimoxazole Rifampicin Azithromycin		Ciprofloxacin 300 mg × 3/day Cotrimoxazole 110 mg × 3/day Rifampicin 450 mg Azithromycin 350 mg	Ciprofloxacin two weeks Cotrimoxazole two weeks Rifampicin 4 weeks Azithromycin 4 weeks
Schuster et al. [102]	Neurologic CSD with hyperactivity	Doxycycline Rifampin		NA	2 weeks
Rosas et al. [176]	encephalitis associated with left arm flaccid paralysis	Doxycycline		100 mg × 2/zi	2 weeks
Bilawsky et al. [179]	Neuroretinitis in pregnant woman	None			
Celiker et al. [147]	Neuroretinitis in three patients	Doxycycline		NA	NA
Raihan et al. [177]	Neuroretinitis in four patients	Azithromycin (3 cases) Doxycycline (1 case)		Azithromycin 250 mg Doxycycline 200 mg	Azithromycin 4–6 weeks Doxycycline 4 weeks
Mutucumarana et al. [120]	VII-th nerve palsy	Azithromycin and rifampin		NA	2 weeks
Zakhour et al. [178]	Transverse myelitis and Guillain-Barré syndrome	Ceftriaxone, vancomycin, doxycycline			Ceftriaxone and vancomycin a few days; Doxycycline 2 weeks
Fouch et al. [109]	Disseminated *Bartonella henselae*	Cephalexin		NA	
Farooque et al. [107]	Persistent focal seizures and encephalopathy	NA			4 weeks
Pinto et al. [180]	aseptic meningitis and neuroretinitis	Azythromicin, Doxycicline, Rifampin		Azythromicin 500 mg; Doxycicline 100 mg; Rifampin 300 mg	Azythromicin a few days; Doxycicline and Rifampin a month

This disease is especially concerning when it occurs in pregnancy because the course of cat-scratch disease is longer and frequently febrile, lasting weeks or months. *B. henselae* DNA was found in postmortem cells of a neonate delivered by a mother who had been diagnosed with *B. henselae* infection; the baby died nine days after birth. These findings indicate that *B. henselae* infection could be detrimental to the fetus, explicitly or implicitly, and might even present a danger of horizontal transmission [89].

## 9. Conclusions

By and large, medical information concerning neurobartonellosis is restricted to case reports and case series, and a clear molecular mechanism behind the emergence of symptoms and pathogenesis of the lesions has not been discovered.

Neurological and ophthalmological symptoms and syndromes can develop at the same time. The period between the beginning of CSD and the commencement of encephalopathy is between days and two months. Encephalopathy is often signaled by disorientation, which might be followed by a decreased state of awareness, varying from sleepiness to a comatose state. Around 50% of individuals have seizures, which vary from localized, self-limited clonic convulsions to status epilepticus. Aphasia, hemiparesis, cranial nerve palsy, and ataxia are all examples of focal neurological symptoms that occur in a minority of individuals. Despite *Bartonella henselae* infrequently generating encephalopathy and meningitis, a favorable prognosis following antibiotic therapy requires clinicians to consider them in routine practice.

Infected individuals may have a variety of visual symptoms of *Bartonella* infections. Neuroretinitis, optic neuritis, focal retinitis, choroiditis, chorioretinitis, exudative maculopathy, serous retinal detachment, and vitritis have all been described in the literature. Other ocular issues, such as branch retinal artery blockage, macular holes, and peripapillary angiomatosis, have also been reported. There have also been reports of conjunctival symptoms such as Parinaud’s oculoglandular syndrome and nonspecific follicular conjunctivitis. 

*B. henselae* is the most often identified cause of neuroretinitis, with about two-thirds of patients exhibiting serologic evidence of prior *B. henselae* infection. Patients with ocular bartonellosis may also present with a variety of posterior segment symptoms, such as posterior uveitis, neuroretinitis, optic neuritis/papillitis, optic disc granuloma, retinal vasculitis, retinal venous or arteriolar occlusions, angiomatosis, acute multifocal inner retinitis or retinal white-dot syndrome and/or retinal infiltrates. 

Whenever a patient presents with injected conjunctivae and granulomatous follicular conjunctivitis, the possibility of Parinaud’s oculoglandular syndrome must be considered, especially if the patient has indicated interaction with cats or kittens. 

## Figures and Tables

**Figure 1 brainsci-12-00217-f001:**
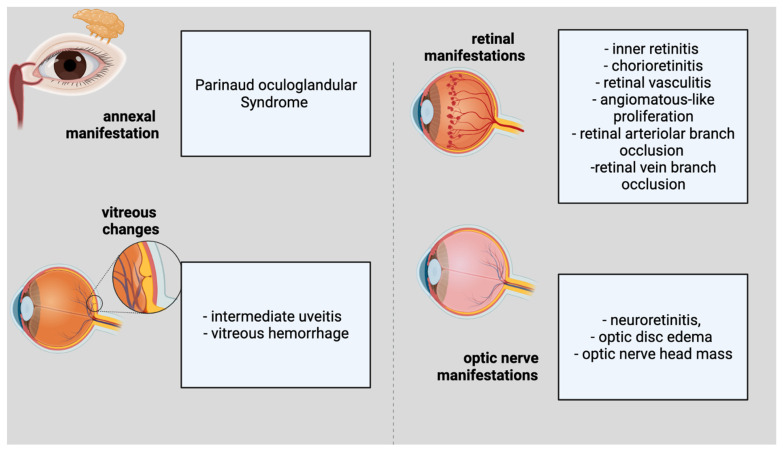
Ophthalmological features of cat scratch disease.

**Table 1 brainsci-12-00217-t001:** Possible diseases triggered by *Bartonella* spp.

Disease	Possible Mechanism	Source
Cat-scratch disease	*B henslae* rapidly infects human erythrocytes and can be found in lymph nodes or affect any cell or organ in the body.	*Bartonella henselae* transmitted by cats or dogs
Carrion’s disease	The bacterium adheres to erythrocyte surfaces. Through bacterial invasion and reproduction, many erythrocytes in the bloodstream are destroyed prematurely, leading to hemolytic anemia.	*Bartonella bacilliformis* transmitted by the night-biting sand fly known as Lutzomyia (formerly Phlebotomus).
Trench fever	Fever is the predominant symptom, with isolated febrile episodes or four-to-five-day feverish episodes or two-to-six-week persistent febrile episodes.	*Bartonella quintana* is transmitted by contamination of a skin abrasion or louse-bite wound with the feces of an infected body louse (Pediculus humanus corporis).

**Table 2 brainsci-12-00217-t002:** Symptoms and signs of CSD.

Symptoms	Signs
Regional pain or body aches	Primary skin lesion that starts as a vesicle
Lymph nodes near the original scratch or bite can become swollen, tender, or painful	Regional unilateral lymphadenopathy
Prolonged fever	Rash
Fatigue	Lack of energy and tiredness
Loss of appetite	Weight loss
Sore throat	Regional signs of inflammation
Abdominal pain	Hepatomegaly and splenomegaly
Headaches	Encephalopathy
Joint pain	Unusual gait

**Table 3 brainsci-12-00217-t003:** Causes of manifestations of *Bartonella henselae* infection.

Inflammatory	Vascular	Neurogenic
Endothelial cells and CD-34 hematopoietic progenitors *Bartonella* adhesin A facilitates attachment to extracellular matrix and mammalian host cells The type IV secretion system VirB/VirD4 is a critical virulence factor Trw-system, extra adhesins, and maybe filamentous hemagglutinins of alpha-, beta-, and gamma-proteobacteria are virulence factors	Arterial infections and vasoproliferative lesions *B. henselae* is quickly absorbed by endothelial cells in vitro, a mechanism mediated by actin *B. henselae* is quickly absorbed by endothelial cells in vitro, a mechanism mediated by actin *B. henselae* stimulates endothelial cell growth through VEGF release Bacterial proliferation results in the release of proinflammatory chemicals and growth regulators, and the cessation of apoptosis, which manifests as new lumps inside the vascular system	Circulating antibodies may induce an immune response by specifically damaging the blood–brain barrier Circulating pathogenic antibodies could enter the nervous system through the blood–brain barrier *B. henselae* may invade human brain vascular pericytes

## Data Availability

No new data were created or analyzed in this study.

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
