# Peer review of "The Clinical Profile of Cat-Scratch Disease’s Neuro-Ophthalmological Effects"

_brainsci, 2022, doi:10.3390/brainsci12020217_

Round 1

Reviewer 1 Report

Comments and Suggestions for Authors

L:42 What could be the explanation of the seasonality of serology?  (or did you think of seasonality of prevalence)

L:59 What will happen with an infected cats/dogs? Are they asymptomatic carriers? Are there any preventive possibilities? eg  vaccinators?

L64: What is Ctenocephalides felis? Take the explanation here from L114-115

L115: What are the vectors other than cat fleas?

L148: Why negative is Bartonella PCR in blood? What is sensitivity/specificity of the Bartonella serology? What is the most specific test for Bartonella?

Figure1 Parinaud syndrome or Parinaud oculo-glandular syndrome?

Please construct a table that summarizes the possible diseases triggered by Bartonella (disease/possible mechanism/ source)

Please construct a table that summarizes the most important general symptoms and signs, in addition to pathognomic symptoms and signs of ophthalmic disease. When should we think of CSD?

Can you show a short case/ picture report about an own case?

Author Response

Thank you very much for your help and support!

Reviewer 1

L:42 What could be the explanation of the seasonality of serology?  (or did you think of seasonality of prevalence)

We have added new information:  LINES 325-373

“The explanation for the seasonal serology is due to both cat behavior and the C. felis life cycle [1]. Adult cat fleas feed on the blood of the host cat and spread B. henselae. Fleas have a four-stage life span. Humidity and temperature are key factors in flea reproduction, development, and lifespan. The seropositivity of B. henselae is significantly greater in cats in warm, humid areas compared with cold, dry climates, owing to the higher prevalence of C. felis fleas in warmer areas [10,11]. As a consequence, cats have a greater number of fleas throughout the summer and fall months than they do in the spring and winter. Sexual behavior in cats can potentially have an effect on the seasonality of CSD [10,11]. Cat reproduction is more common in spring and summer, and kittens remain with their mothers until the age of 3–4 months. Additionally, during the fall, people are more in-clined to get kittens. B. henselae infection seems to be more prevalent in young cats, and the infection rates tend to fall with the duration of cat ownership [12]. Additionally, cats encounter more fleas throughout the summer and fall, which facilitates the transmission of B. henselae from cat to cat.”

L:59 What will happen with an infected cats/dogs? Are they asymptomatic carriers? Are there any preventive possibilities? eg  vaccinators?

We have added new information:  LINES: 174-295

“Cats and dogs are usually asymptomatic carriers. Diseases of cats infected with B. henselae can include anemia and diaphragmatic myositis, and markers and manifestations of infection/inflammation, including eosinophilia, fever, hyperglobulinemia, lethargy, and lymphadenomegaly, are also present. In addition, mild neurological signs may be present, as well as cardiac manifestations such as pyogranulomatous myocarditis, en-docarditis, endomyocarditis, and endocardial fibrosis complex. Ocular manifestations include uveitis, conjunctivitis, keratitis, and corneal ulcers [7].

The manifestations of the infection of B. henselae in dogs include lymphadenomegaly, endocarditis, eosinophilia, epistaxis, fever, and granulomatous inflammation [6,7]. Glucidic metabolism can be affected by hyperinsulinemic and hypoglycemia syndrome. The hepatic manifestations include granulomatous hepatitis and peliosis hepatis [6,7]. Additionally, vasoproliferative lesions can be present.

The primary approach to prevention would be to avoid contact with cats. However, cat owners can adopt the following highly effective preventative techniques for flea management in cats: washing hands thoroughly after interaction with a cat; avoiding contact with stray cats—especially kittens; avoiding cat licks, particularly in the mouth, nose, and ophthalmic region; and generally maintaining a pleasant relationship with the cat, without scratching or biting.

A key strategy for preventing B. henselae infection in cats is to manage fleas and other ectoparasites. Additionally, keeping cats’ claws short is beneficial. With regards to the surroundings, owners should maintain cleanliness and be vigilant about pest man-agement. Additionally, minimizing a cat’s interaction with other or stray cats and po-tentially infected animals may be beneficial. Owners should also schedule routine vet-erinarian health examinations. There is currently no vaccine available to prevent infection with Bartonella.”

L64: What is Ctenocephalides felis? Take the explanation here from L114-115

We have added new information: Lines: 394-400

“Ctenocephalides felis spreads Bartonella henselae between cats and frequently between cats and humans through infected flea excrement, resulting in an initial inoculation lesion and lymphadenopathy resulting from the incorporation of flea excrement from Ctenocephalides felis in skin abrasions caused by cat scratches or bites [18,19]. Numerous Bartonella sub-species have been linked to human illnesses, but Bartonella henselae seems to be the most often involved with ophthalmic inflammations. “

L115: What are the vectors other than cat fleas?

We have added new information: Lines: 295-300:  

“Ectoparasites such as fleas, ticks, and mites are often discovered on  cats and dogs and are capable of harboring Bartonella spp. Lice, fleas, and sandflies have been identified as vectors of five Bartonella species including Bartonella henselae. Numerous mites, keds, and biting flies, as well as probably ticks, are now considered possible or potential vectors of Bartonella transmission.”

L148: Why negative is Bartonella PCR in blood? What is sensitivity/specificity of the Bartonella serology? What is the most specific test for Bartonella?

We have added new information: Lines: 502-557:

 “Regrettably, no gold standard for conclusive CSD diagnosis has been developed. Con-sidering the technical difficulties associated with isolating B. henselae from patient specimens, serology seems to have become the gold standard for diagnosing CSD. This is typically accomplished using available indirect immunofluorescence assays (IFAs) ca-pable of detecting IgM and IgG antibodies to B. henselae [24–26]. Because of its high sensitivity and specificity, real-time polymerase chain reaction (PCR) on lymph nodes or other clinical samples has been recommended as a viable approach for detecting B. henselae DNA in suspected cases of CSD [24–26]. However, this approach is restricted by the necessity for surgical collection by lymphadenectomy or biopsy, which could be addressed by conducting real-time PCR on DNA samples obtained from extracted pus or blood. Even so, this method might not be indicated in patients lacking bacterial DNAemia.

The direct identification of B. henselae using microbiological cultures is difficult owing to the bacterium’s slow growth rate. Although more sensitive than microbiological cul-ture, serological analysis for anti-B. henselae IgM and IgG antibodies using IFA, the first-line diagnostic test for CSD, lacks specificity because of the seropositivity of many asymptomatic people due to past animal exposure [24–26]. Additionally, the discovery of anti-B. henselae IgM antibodies is a marker of acute illness that persists in the blood only for about three months after exposure. B. henselae IgG antibodies may be identified in the blood for up to 5–7 months after exposure, with only 25% of individuals maintaining IgG seropositivity after one year [24–26].”

Figure1 Parinaud syndrome or Parinaud oculo-glandular syndrome?

Done. The figure was modified.

Please construct a table that summarizes the possible diseases triggered by Bartonella (disease/possible mechanism/ source)

The table was created and added in the text: LINE 165.

Disease

Possible mechanism

Source

Cat-scratch disease

B henslae rapidly infects human erythrocytes and can be found in lymph nodes or affect any cell or organ in the body.

Bartonella henselae transmitted by cats or dogs

Carrion’s disease

The bacterium adheres to erythrocyte surfaces. Through bacterial invasion and reproduction, many erythrocytes in the bloodstream are destroyed prematurely, leading to hemolytic anemia.

Bartonella bacilliformis transmitted by the night-biting sand fly known as Lutzomyia (formerly Phlebotomus).

Trench fever

Fever is the predominant symptom, with isolated febrile episode or four-to-five-day feverish episodes or two-to-six-week persistent febrile episodes.

Bartonella quintana is transmitted by contamination of a skin abrasion or louse-bite wound with the faeces of an infected body louse (Pediculus humanus corporis).

Please construct a table that summarizes the most important general symptoms and signs, in addition to pathognomic symptoms and signs of ophthalmic disease. When should we think of CSD?

LINE 499: The table and the text were added in the manuscript.

Symptoms

Signs

Regional pain or body aches

Primary skin lesion that starts as a vesicle

Lymph nodes near the original scratch or bite can become swollen, tender, or painful

Regional unilateral lymphadenopathy

Prolonged fever

Rash

Fatigue

Lack of energy and tiredness

Loss of appetite

Weight loss

Sore throat

Regional signs of inflammation

Abdominal pain

Hepatomegaly and splenomegaly

Headaches

Encephalopathy

Joint pain

Unusual gait

LINES: 490-498

In the case of fever of unknown etiology or prolonged fever and lymphadenopathy, patients should be asked whether they have been in contact with a cat or dog, including if they have been scratched or bitten, and if they have acquired a primary skin lesion that started as a vesicle at the inoculation site. While prior exposure to cats is helpful for di-agnosis, it is not required to establish a diagnosis. Additionally, in the case of manifes-tations such as pain, malaise, and anorexia, as well as a low-grade fever, musculoskeletal manifestations, and hepatosplenomegaly, cat-scratch disease might be suspected.

Can you show a short case/ picture report about an own case?

We have a patient with neuro-ophthalmic symptoms of cat scratch disease, who inspired us to do this article, but unfortunately we cannot add the case report because the patient in question does not want to be part of literature studies.

Reviewer 2 Report

Comments and Suggestions for Authors

General comments

Whilst I applaud the authors for their painstaking review of the literature, the article at present is rather disjointed and the information difficult to assimilate for a reader not as conversant with the subject as you will be. I feel it needs significantly shortening and this can be achieved by combining much of the information in the discussion that is new in the relevant sections as there appears to be much revisiting of topics here - indeed a review does not have to have a discussion.

The conclusion should be a short summary of what this article adds - there should not be new references within it.

There are a number of subtle errors of spelling/meaning e.g. line 409 neurotoxic should be neurotoxin. Please look carefully at the manuscript to correct these.

Specific comments

There is a tendency to introduce acronyms without an initial explanation (VEGF, LAP)

In line 259/260 you state two percentages for the incidence of neurological events (problems 2% - signs 7%) - the requires clarity of explanation as to why the figures are different please.

Figure 1 - title should really be ophthalmological manifestations of CSD - most of these conditions are not neuro-ophthalmic. 

Line 427 - please specify exactly which investigation you are quoting with your statement " cell culture technique used in this investigation"

Line 492 and 503  - please do not use colloquialisms - "kid" should be "child" How old is a "youngster" - line 499 and 508? use well defined terms or give the age in a specific case.

Lines 677/8 [148]. "Although B. henselae is the most often encountered infectious cause of neuroretinitis, therapy continues to persist owing to the disease's self-limiting character." this doesn't make sense - please find another way of saying what you mean here.

Author Response

Thank you very much for your help and support!

Reviewer 2

General comments

Whilst I applaud the authors for their painstaking review of the literature, the article at present is rather disjointed and the information difficult to assimilate for a reader not as conversant with the subject as you will be. I feel it needs significantly shortening and this can be achieved by combining much of the information in the discussion that is new in the relevant sections as there appears to be much revisiting of topics here - indeed a review does not have to have a discussion.

The conclusion should be a short summary of what this article adds - there should not be new references within it.

We have modified the conclusion accordingly.

There are a number of subtle errors of spelling/meaning e.g. line 409 neurotoxic should be neurotoxin. Please look carefully at the manuscript to correct these.

We have made extensive english grammar and spelling modifications through the MDPI EDITING SERVICE.

Specific comments

There is a tendency to introduce acronyms without an initial explanation (VEGF, LAP)

The explanations were added for every acronym.

In line 259/260 you state two percentages for the incidence of neurological events (problems 2% - signs 7%) - the requires clarity of explanation as to why the figures are different please.

Done. The phrase was modified. Lines: 1082-1084

“Neurologic complications from B. henselae infection are infrequent, occurring in around 2–7% of infected persons. Neurological symptoms often manifest two weeks after the onset of fever and lymphadenopathy [87,88].”

Figure 1 - title should really be ophthalmological manifestations of CSD - most of these conditions are not neuro-ophthalmic. 

Done. The title was modified accordingly.

Line 427 - please specify exactly which investigation you are quoting with your statement " cell culture technique used in this investigation"

The phrase was modified.

Line 492 and 503  - please do not use colloquialisms - "kid" should be "child" How old is a "youngster" - line 499 and 508? use well defined terms or give the age in a specific case.

The modifications were made.

Lines 677/8 [148]. "Although B. henselae is the most often encountered infectious cause of neuroretinitis, therapy continues to persist owing to the disease's self-limiting character." this doesn't make sense - please find another way of saying what you mean here.

The lines were modified. LINES: 2036-2041

“While B. henselae seems to be the most frequently observed infectious cause of neuroretinitis, treatment persists due to the disease’s self-limiting nature. Thus, according to two specialists, the rarity of Bartonella neuroretinitis complicates treatment [166]. Lee et al. believe that antimicrobial treatment reduces the duration of the disease symptoms and that, therefore, it should be treated with antibiotics until the laboratory testing is com-pleted [167].”
